# Prevalence and Associated Factors for Purchasing Antibiotics Without a Prescription Among Patients in Rural South Africa: Implications for Addressing Antimicrobial Resistance

**DOI:** 10.3390/antibiotics14121273

**Published:** 2025-12-16

**Authors:** Tiyani Milta Maluleke, Morgan Tiyiselani Maluleke, Nishana Ramdas, Ana Golić Jelić, Amanj Kurdi, Audrey Chigome, Stephen M. Campbell, Vanda Marković-Peković, Natalie Schellack, Brian Godman, Johanna C. Meyer

**Affiliations:** 1Department of Public Health Pharmacy and Management, School of Pharmacy, Sefako Makgatho Health Sciences University, Molotlegi Street, Garankuwa, Pretoria 0208, South Africa; tiyanim@gmail.com (T.M.M.); nishanaramdas@gmail.com (N.R.); audrey.chigome@smu.ac.za (A.C.); stephen.campbell@smu.ac.za (S.M.C.); hannelie.meyer@smu.ac.za (J.C.M.); 2Saselamani Pharmacy, Saselamani 0928, South Africa; malulekemorgan68@gmail.com; 3Department of Pharmacy, Faculty of Medicine, University of Banja Luka, 78000 Banja Luka, Republic of Srpska, Bosnia and Herzegovina; ana.golic@med.unibl.org (A.G.J.); vanda.markovic-pekovic@med.unibl.org (V.M.-P.); 4Strathclyde Institute of Pharmacy and Biomedical Sciences, Strathclyde University, 161 Cathedral Street, Glasgow G4 0RE, UK; amanj.baker@strath.ac.uk (A.K.); 5College of Pharmacy, Hawler Medical University, Erbil 44001, Kurdistan Region, Iraq; 6College of Pharmacy, Al-Kitab University, Kirkuk 36015, Iraq; 7South African Vaccination and Immunisation Centre, Sefako Makgatho Health Sciences University, Molotlegi Street, Garankuwa, Pretoria 0208, South Africa; 8School of Health Sciences, University of Manchester, Manchester M13 9PL, UK; 9Department of Pharmacology, Faculty of Health Sciences, University of Pretoria, Pretoria 0084, South Africa; natalie.schellack@up.ac.za; 10Antibiotic Policy Group, City St. George’s, University of London, London SW17 0RE, UK

**Keywords:** antibiotics, antimicrobial resistance, antimicrobial stewardship, health policy, patients, prescribers, prevalence antibiotic use, self-purchasing antibiotics, South Africa

## Abstract

**Background:** Antimicrobial resistance (AMR) is now a critical issue in South Africa, enhanced by considerable inappropriate prescribing of antibiotics. There is currently variable dispensing of antibiotics without a prescription. Where this occurs, it is principally for urinary tract infections (UTIs) and sexually transmitted infections (STIs). Consequently, there is a need to comprehensively evaluate antibiotic dispensing patterns and factors influencing this to reduce AMR. **Methods:** A previously piloted questionnaire was administered to patients exiting three different categories of community pharmacies in a rural province. The questionnaire included data on the prevalence of antibiotics dispensed, whether without a prescription, and the rationale for this. **Results:** A total of 465 patients leaving community pharmacies with a medicine were interviewed. 54.4% of interviewed patients were dispensed at least one antibiotic, with 78.7% dispensed these without a prescription from either independent or franchise pharmacies. Metronidazole (36.1%) and azithromycin (32.7%) were the most dispensed antibiotics. STIs were the most common infectious disease for which an antibiotic was dispensed (60.1%), with 99.6% dispensed without a prescription. Upper respiratory tract infections (URTIs) were the most common infection where antibiotics were dispensed with a prescription (60.0%), with little dispensing without a prescription (7.1%). The most frequently cited reasons for obtaining antibiotics without a prescription were prior use (56.8%), long waiting times at PHC clinics (15.6%), and financial constraints (6.0%). **Conclusions:** There is an urgent need to review community pharmacists’ scope of practice, including allowing them to prescribe antibiotics for infectious diseases such as UTIs, similar to other countries. Concomitantly, utilise trained community pharmacists to engage with prescribers to improve future antibiotic use, especially for URTIs.

## 1. Introduction

Antimicrobial resistance (AMR) is recognised as a leading public health challenge globally [1,2,3,4], with bacterial infections now the second leading cause of death worldwide [2]. Published estimates suggest there were up to 5 million deaths annually associated with bacterial resistance in recent years, and the number continuing to rise [5,6,7]. The greatest mortality is currently seen among low- and middle-income countries (LMICs), which include African countries [6,7,8,9,10]. This includes children across Africa and Asia [11]. Alongside this, there are considerable cost implications associated with AMR among LMICs [12,13,14].

A key driver of AMR is inappropriate antibiotic utilisation in humans, enhanced by global antibiotic consumption increasing by 65% from 2000 to 2015, driven principally by their increased utilisation in LMICs [5,7,15,16,17]. This increased utilisation among LMICs, coupled with the growing use of Watch and Reserve antibiotics with their greater resistance potential as classified by the World Health Organization’s (WHO) AWaRe system, continues to increase AMR [5,7,11,18,19,20,21,22]. High levels of AMR have been exacerbated by no change in the considerable levels of inappropriate prescribing of antibiotics, or their inappropriate dispensing without a prescription, among LMICs in recent years [23,24].

South Africa is no exception, with high rates of AMR, which are exacerbated by increasing utilisation of antibiotics, with their utilisation increasing by 50% in the public sector between 2018 and 2022 [25,26,27,28,29]. High rates of AMR are also enhanced by a decrease in the use of Access antibiotics during this period from 79% of total antibiotic use to 48% [29]. Alongside this, there has been a rapid increase in the use of Reserve antibiotics in recent years, with their greater resistance potential, coupled with concerns with prescriber adherence to current antibiotic guidelines in South Africa, all adding to AMR [22,28,29,30]. Concerns with current prescribing of antibiotics in primary healthcare (PHC) clinics have continued (Appendix A), which is similar to the situation among LMICs [23]. As a result, there have been urgent calls to the Ministry of Health in South Africa to increase its focus on AMR as a critical public health issue alongside increased antimicrobial stewardship (AMS) activities [31,32].

Among LMICs, especially among African countries, there is an urgent need to reverse current high levels of inappropriate dispensing of antibiotics without a prescription for essentially self-limiting infections, which often involves Watch and Reserve antibiotics, further exacerbating AMR [33,34,35,36,37,38,39,40,41]. These high rates of purchasing of antibiotics without a prescription are driven by a number of key factors [34,37,40,41]. These include the accessibility of community pharmacies versus often long waiting times and costs to see healthcare professionals (HCPs) in PHC clinics, high co-payments, beliefs that their infectious diseases are minor and self-manageable, patient demand including their previous experiences with antibiotics, their lack of knowledge regarding antibiotics and AMR, as well as profit motives among pharmacists [24,34,40,42,43,44,45,46,47]. These persistent factors have resulted in little change in the purchasing of antibiotics without a prescription across LMICs in recent years [24].

Alongside this, the lack of knowledge among patients and carers in LMICs regarding antibiotics, AMR, and AMS is also a concern. This is because of the increasing recognition of their key role in influencing both the prescribing and dispensing of antibiotics in primary care among LMICs (Appendix A). This includes for self-limiting conditions such as upper respiratory tract infections (URTIs) [48,49,50,51]. Addressing these concerns is critical to reducing AMR in LMICs, especially with antibiotic utilisation in primary care currently accounting for up to 90% of total antibiotic use in humans [52,53].

Previous research in South Africa has shown contradictory evidence regarding the extent of purchasing of antibiotics without a prescription, including among rural provinces, despite this being illegal in the country (Table 1) [54,55]. However, where this does occur, the purchasing of antibiotics without a prescription has been more likely from independent pharmacies [55]. The differences in results between the published studies (Table 1) may reflect differences in populations and context, availability of and costs associated with consulting with HCPs in PHC centres, as well as changing economic circumstances. We have seen, for instance, growing rates of the purchase of antibiotics without a prescription in Zimbabwe following the worsening of the economic situation in the country [34].

As a result of the current concerns with antibiotic use in primary care in South Africa, including both the prescribing and dispensing of antibiotics (Appendix A and Table 1), there is an urgent need to build on recent studies to improve future antibiotic use in the country [29,31,60,61]. This includes understanding the extent to which, and why, patients purchase antibiotics without a prescription in South Africa and the nature of these antibiotics in terms of their AWaRe classification. Our motivation for this is the acknowledgement that community pharmacists and their assistants may under-report dispensing of antibiotics without a prescription, especially if such activities are currently illegal [55,62].

The study described in this paper was part of a larger mixed-methods study to investigate the extent of purchasing of antibiotics without a prescription from community pharmacies in a rural province in South Africa (Table 1) [60]. In view of this, the objectives of this part of the study conducted among patients were firstly to ascertain whether they had been dispensed antibiotics, and if so, which antibiotic and for what clinical indication. Secondly, to ascertain whether any of the dispensed antibiotics had been purchased without a prescription, and the reasoning behind this. Alongside this, the indication for any antibiotic being dispensed without a prescription, i.e., whether for a self-limiting condition such as URTIs, which is a common condition for purchasing antibiotics without a prescription across LMICs (Appendix A), or for a sexually transmitted infection (STI), as seen in previous studies in South Africa (Table 1) [37,54,58,63]. Lastly, to explore whether there was any association between the category of community pharmacy and the extent of purchasing of antibiotics without a prescription, as seen in previous studies (Table 1). The findings will inform future activities and interventions to help reduce AMR, given current concerns in South Africa [31].

To the best of our knowledge, we believe this is the first study to comprehensively evaluate patterns of antibiotic dispensing, including their reported indication for use, the extent of purchasing of antibiotics without a prescription, and the contextual factors influencing this in a rural South African province. The inclusion of a substantial patient cohort in this province should allow for granular assessment of dispensing practices, including the dispensing of antibiotics without a prescription and pharmacy typologies, generating evidence to inform future targeted AMS activities in South Africa and wider.

## 2. Results

### 2.1. Sample Size and Response Rate

A total of 701 patients were approached for possible participation during the study. Of these, 87 patients were not carrying any medication in their medicine or pharmacy bags; consequently, they were excluded as they did not meet the inclusion criteria. Of the remaining 614 patients, 149 patients declined to participate, citing various reasons. Ultimately, 465 interviews with complete responses were obtained, resulting in an overall response rate of 75.7%.

### 2.2. Socio-Demographic Characteristics of Patients

The majority of patients interviewed were exiting independent pharmacies (269; 57.8%) (Table 2), with the remainder exiting from either franchise (24.9%) or chain pharmacies (17.2%). This aligned with the distribution of the different pharmacy categories in this rural province (see Section 4.2. Target Population and Study Sample).

The mean (39.7 years; SD: 13.0) and median (39 years; IOR: 18) age of the participants were similar, with equal distribution between males and females (Table 2). The most commonly preferred language of communication was English (45.8%), with the vast majority of participants (87.5%) having a tertiary qualification in addition to having completed secondary school.

### 2.3. Medicine Items Dispensed to Patients and/or Purchased by Them from Community Pharmacies

Amongst the interviewed patients, 127 (27.3%) were dispensed medicines with a prescription, while 338 (72.7%) patients obtained their medication without a prescription, which included over-the-counter (OTC) medicines typically used for minor ailments. In total, 1079 medicine items were supplied to the surveyed patients, of which 369 (34.2%) were dispensed with a prescription. Table 3 provides more details about the distribution of medicine items dispensed to patients with or without a prescription.

Of the 1079 medicine items dispensed, 377 items were antibiotics, of which 66 (17.5%) were dispensed to patients with a prescription, while 311 (82.5%) antibiotic items were dispensed without a prescription.

### 2.4. Antibiotics Dispensed to Patients With or Without a Prescription

More than half (54.4%; 253/465) of the interviewed patients were dispensed at least one antibiotic, of whom only 54 patients (21.3%) received their antibiotics following a prescription, while the majority (78.7%; 199/253) of patients were dispensed their antibiotics without a prescription.

Table 4 shows that the majority of patients who received an antibiotic were dispensed only one antibiotic (52.2%; 132/253). Appreciably more patients who were dispensed two or more antibiotics were dispensed these without a prescription. Adults accounted for the majority of dispensed antibiotics (68.8%; 174/253), as opposed to antibiotics destined for children (31.2%; 79/253). Adults also accounted for a greater proportion among patients who were dispensed antibiotics without a prescription (70.9%; 141/199).

### 2.5. Antibiotics Dispensed to Patients from Different Categories of Pharmacies

Independent pharmacies accounted for the majority of patients being dispensed antibiotics (63.3%; 161/253) compared with franchise pharmacies (25.7%; 65/253) and chain pharmacies (10.7%; 27/253). Of the patients who were dispensed antibiotics without a prescription, most patients had these dispensed from either independent (74.4%; 148/199) or franchise pharmacies (25.6%; 51/199), with no chain pharmacy dispensing antibiotics without a prescription (Appendix A).

Overall, 91.9% (148/161) of patients visiting independent pharmacies had their antibiotics dispensed without a prescription. Similar to the situation with independent pharmacies, 78.5% (51/65) of patients visiting franchise pharmacies were dispensed their antibiotics without a prescription (Appendix A). The differences in terms of antibiotic dispensing with or without a prescription between the different pharmacy categories were statistically significant (*p* < 0.001) with a moderate association (Cramer’s V = 0.405).

### 2.6. Types of Antibiotics Dispensed to Patients Distributed by the AWaRe Classification

A total of 377 antibiotics were dispensed among the 253 patients with or without a prescription (Table 5). The most frequently dispensed antibiotics were metronidazole (36.2%; 136/377) and azithromycin (32.7%; 123/377), accounting for nearly 69% of all antibiotics dispensed. Of all the antibiotics dispensed without a prescription, metronidazole (42.4%; 132/311) and azithromycin (33.4%; 104/311) were again the most frequently dispensed. Of all antibiotics dispensed with a prescription, azithromycin (28.8%; 19/66) and co-amoxiclav (27.3%; 17/66) were the most common antibiotics dispensed (Table 5). Differences between antibiotics dispensed with and without a prescription were statistically significant (*p* < 0.001) with a strong association (Cramer’s V = 0.831).

Of the 376 antibiotic items dispensed, not including metronidazole gel, 56.4% (212/376) belonged to the Access group, with 43.6% (164/376) belonging to the Watch group and none to the Reserve group (Table 5). Among the antibiotics dispensed with a prescription (excluding metronidazole gel), 63.1% (41/65) were Access antibiotics and 36.9% (24/65) were Watch, with no Reserve antibiotics dispensed. Among the items dispensed without a prescription, 55.0% (171/311) were Access antibiotics, and 45.0% (140/311) were Watch antibiotics (Appendix A). However, the differences between WHO AWaRe categories and their dispensing with or without a prescription were not statistically significant (*p* = 0.290; OR = 1.399 [95% CI: 0.806; 2.427]) with a weak association (Cramer’s V = 0.0617).

### 2.7. Indications or Conditions for Which Patients Sought Treatment

STIs were the most common perceived infectious disease specified by participating patients for which they sought treatment and for which an antibiotic was dispensed, accounting for 60.1% (226/377) of all antibiotics dispensed (Table 6).

Notably, 99.6% of STI-related antibiotics (225/226) were dispensed without a prescription, with STIs overall accounting for 72.3% (225/311) of the infectious diseases where an antibiotic was dispensed without a prescription.

URTIs were the most common perceived infectious disease for which antibiotics were dispensed with a prescription (60.0%; 39/65), with very limited dispensing of antibiotics without a prescription for patients with URTIs (7.1%; 3/42). The differences regarding indications for which an antibiotic was dispensed with or without a prescription were statistically significant (*p* < 0.001) with a strong association (Cramer’s V = 0.841).

### 2.8. Patients’ Reasons for Accessing Community Pharmacies

Among the 465 patients interviewed leaving community pharmacies across the province, convenience was the most frequently cited reason for choosing a particular community pharmacy (31.8%; 148/465), principally relating to patients collecting medicines, including antibiotics, with a prescription or OTC (71.6%; 106/148) (Table 7). Easy parking was also a key issue when patients chose a particular pharmacy for collecting medicines (14.6%; 68/465), as indicated by the majority (88.2%; 60/68) who were dispensed their medicines, including antibiotics, with a prescription or OTC. In contrast, familiarity with staff was the dominant reason among those seeking antibiotics without a prescription (55.8%; 111/199), followed by convenience (21.1%; 42/199).

Among the 199 patients who were dispensed antibiotics without a prescription, the most frequently cited reason for obtaining these was prior use of the same antibiotic (56.8%; 113/199). Other notable reasons included long waiting times at PHC clinics (15.6%; 31/199) and financial constraints (6.0%; 12/199), potentially associated with the costs incurred when going to PHC clinics and currently limited finances. Table 8 provides more details on patients’ reasons for obtaining an antibiotic without a prescription, which builds on the reasons for choosing a particular pharmacy as shown in Table 7.

## 3. Discussion and Next Steps

We believe this is the most extensive study that has been undertaken among patients in a rural province in South Africa to assess current patterns of the dispensing of antibiotics. This includes the indications or conditions for which patients sought treatment, the extent of any purchasing of antibiotics without a prescription, and the rationale for this activity.

Whilst this was not a prescription review, antibiotics made up an appreciable portion (34.9%) of the medicines dispensed, with over half of the patients (54.4%) who exited community pharmacies with a medicine, including OTC medicines, being dispensed at least one antibiotic. These rates are consistent with national patterns, including the review by Chigome et al. (2023), where antibiotics constituted between 52.9% and 78% of all prescriptions among PHC clinics in South Africa [30]. In addition, a recent point prevalence survey among PHC clinics in South Africa where antibiotics were prescribed to 87% of patients [61]. This suggests appreciable overprescribing and dispensing of antibiotics, especially when nearly half of all patients dispensed antibiotics in this study (47.8%) were dispensed more than one type, raising concerns with inappropriate antibiotic combinations or repeated use of broad-spectrum antibiotics.

Another identified concern was that 72.7% of patients were dispensed medicines without a prescription, accounting for 65.8% of all items dispensed. While some of these were traditional OTC medicines, the majority of antibiotic items dispensed (82.7%) were without a prescription, more prevalent among adult patients (81.0%) than for children (73.4%). This contrasts with the findings of Anstey Watkins et al. (2019) and Do et al. (2021), as well as the findings of Maluleke et al. (2025) among community pharmacists in the same rural province (Table 1) [56,57,60]. However, similar to the findings of Mokwele et al. (2022), where UTIs, along with concerns with STIs, were one of the two patient scenarios associated with the purchasing of antibiotics without a prescription (Table 1) [54]. There were also similar high rates in the two pilot studies of Sono et al. (2024 and 2025) involving patients exiting community pharmacies in this rural province (Table 1) [58,59]. We believe the findings would have been very different if we had used simulated patients presenting with acute respiratory infections such as URTIs, which is typically a key indication used in these types of studies [64,65,66,67,68,69].

Similarly to the findings of Mokwele et al. (2022), Sono et al. (2024, 2025), and Maluleke et al. (2025), there was no dispensing of antibiotics without a prescription from chain pharmacies [54,58,59,60]. However, greater dispensing of antibiotics without a prescription was seen among franchise pharmacies in this study compared with the pilot studies of Sono et al. [58,59]. The differences in the extent of purchasing of antibiotics without a prescription between the different pharmacy types may reflect differences in the way antibiotic use is recorded and monitored and the different influences of patients according to pharmacy types (Appendix A). Independent pharmacies, which are often embedded within lower-income communities, may offer more lenient or community-tailored dispensing practices in response to local patient demand. Franchise pharmacies, despite operating within semi-regulated corporate frameworks, may also similarly provide antibiotics based on syndromic cues alongside patient demand. Chain pharmacies, by contrast, exhibited consistent adherence to prescription protocols, reflecting centralised governance and structured staffing policies irrespective of patient demand. These differences were statistically significant and suggest that pharmacy type in South Africa is independently associated with the likelihood of dispensing antibiotics without a prescription, similar to previous studies [54,58,60].

Encouragingly, within community pharmacies, there was very limited dispensing of antibiotics without a prescription for URTIs in this rural province (3.7%), especially when compared with the indications where antibiotics were prescribed. This reflects earlier feedback from community pharmacists and pharmacist assistants in this rural province, where 98.1% of pharmacists and 97.6% of pharmacist assistants indicated they always or mostly offered symptomatic relief before dispensing antibiotics without a prescription to patients with self-limiting conditions such as colds and influenza [60]. This contrasts with other LMICs, where URTIs are a key condition where antibiotics are often dispensed without a prescription [37,63], as well as prescribers in South Africa (Appendix A).

One key identified issue that needs further research is the extent to which STIs were the principal indication where antibiotics were dispensed without a prescription (59.7% of all indications), similar to the pilot studies [55,59]. This may well reflect prior patient knowledge regarding which antibiotics are effective in this situation, along with patients not wanting to visit PHC clinics to see HCPs, versus the familiarity with personnel within independent and franchise pharmacies. This suggests pharmacies, particularly independent and franchise pharmacies, are stepping into the challenges arising from constrained public health systems, functioning as accessible but sometimes currently unregulated points of care, especially where patients may wish to limit the number of HCPs knowing about their condition. This suggests that community pharmacists and pharmacist assistants have a key future role in educating patients on aspects of prevention among patients repeatedly presenting with STIs, building on previous studies [68,69,70,71,72]. However, further qualitative research is needed before we can say anything with certainty in this rural province. Alongside this, we are aware that qualified community pharmacists are able to dispense certain antibiotics without a prescription for patients with STIs in Ghana, as well as for patients with UTIs, with good results (Appendix A) [65,73,74,75,76]. These provide exemplars for the Government and other key stakeholders in South Africa going forward to improve the management of patients with UTIs and STIs in primary care.

Other key issues and challenges that must be addressed to improve future antibiotic use in South Africa include effectively tackling continuing high use of Watch antibiotics, which may reflect high rates of STIs among surveyed patients [29,31]. This is important given the recent goals from the United Nations General Assembly on AMR, where a target was set of 70% use of Access antibiotics across sectors to reduce AMR [77,78,79]. Greater education among both prescribers and dispensers of the WHO AWaRe system and guidance is one potential way forward [21,80,81]. Increased AWaRe-ness among all key stakeholder groups, including physicians and patients, should also help reduce current high rates of inappropriate prescribing of antibiotics for patients with self-limiting conditions such as URTIs (60.9% of antibiotics dispensed with a prescription). In addition, greater adherence to the WHO AWaRe guidance should reduce the current extensive range of antibiotics being prescribed or dispensed in primary care in this rural province, including without a prescription, as well as the extent of Watch antibiotics [80,81]. Overall, Access antibiotics accounted for only 56.4% of all antibiotics dispensed in this study, well below the 2024 United Nations target of 70% [79].

The extensive range of antibiotics prescribed and dispensed with and without a prescription within the context of a rural South African province, also presents challenges for stock management, cost containment, and forecasting among community pharmacists, unless appropriate systems are in place [82].

To address these combined concerns, AMS efforts should focus on limiting the prescribing and dispensing of key antibiotics to those recommended in the WHO AWaRe guidance [80,81], emphasising those in the Access group, through targeted antimicrobial stewardship programmes (ASPs) involving community pharmacists with other key stakeholder groups (Appendix A).

Another key issue arising from this study is the considerable difference in the extent of dispensing of antibiotics without a prescription stated by community pharmacists and pharmacist assistants in this rural province, at only 8.6% of all antibiotics when these were dispensed (Table 1), versus the feedback from patients in the same locality in this study. The differences may well reflect knowledge among community pharmacists and pharmacist assistants that such practices are currently prohibited in South Africa, and hence why they understated the extent of this activity [60]. We are aware that the findings from simulated patients in the study of Mokwele et al. (2022) also demonstrated appreciably higher rates of self-purchasing when using simulated patients (Table 1), adding to this suggestion, which has been commented on before [54]. These findings should be considered when assessing the findings from self-completed questionnaires from community pharmacy personnel in LMICs without any verification.

Potential activities to improve future antibiotic use among all key stakeholders in South Africa and wider are outlined in Table 9. These are based on the considerable experiences of the co-authors working in infectious diseases across Africa and other LMICs [15,34,37].

While this study had some strengths, we are also aware of a number of limitations. The strengths include the fact that the study was conducted in the same location and approximately the same time, where pharmacists and assistants were surveyed to allow for comparative analysis. This was seen as particularly important given, as mentioned, concerns with their under-reporting of selling antibiotics without a prescription [62]. There was also detailed antibiotic-level analysis incorporating the WHO AWaRe classification and syndromic indications for antibiotics dispensed with and without a prescription.

In terms of limitations, one of the reasons patients requested antibiotics without a prescription was their prior experience using the same antibiotic. However, the study did not ascertain whether any previous antibiotic use was based on a valid prescription or a self-purchase. This distinction is important, as patients who believe they recovered after using a prescribed antibiotic may be more inclined to seek the same medication without a prescription when experiencing similar symptoms. The lack of this contextual detail limits our ability to fully interpret the motivations behind non-prescription antibiotic requests, and we will be exploring this further in future studies.

We are also aware that at the time of the study, some patients were asked by nurses in PHC clinics to purchase specific antibiotics from community pharmacies, as some items were unavailable and they were unable to issue external prescriptions. These patients were classified as paying for their antibiotics, i.e., without a prescription, for ease of analysis. However, the numbers were limited as there was limited dispensing of antibiotics without a prescription for patients with URTIs, suggesting adequate stock levels for a range of antibiotics among PHCs. This suggests that PHC facilites generally maintained adequate antibiotic stock levels, with only a few patients directed to community pharmacies to purchase their suggested antibiotics due to temporary shortages (Table 7).

Finally, we are also aware that we conducted this study in only one rural province in South Africa for the reasons specified. Consequently, the findings may not be fully applicable to other settings in South Africa or other LMICs. However, despite these limitations, we believe overall our findings are robust, providing appreciable insights that can be taken forward by all key stakeholders in South Africa and wider to improve future antibiotic use.

## 4. Materials and Methods

### 4.1. Study Design and Setting

This study is part of a larger mixed-methods study and was conducted as a cross-sectional descriptive survey among patients as they were exiting community pharmacies in a rural province. As previously described [60,94], a rural province of South Africa was purposively selected for this study, as extended travelling distances to PHC facilities, coupled with long waiting times to see HCPs and shortages of physicians in ambulatory care in this province, is a common phenomenon [95]. Consequently, this enhances the potential for patients to approach community pharmacists and their assistants directly for better access to, and potential purchasing of, antibiotics without a prescription [34,55,60].

The study was designed to obtain data directly from patients immediately after they had exited community pharmacies. We deliberately chose this approach versus using simulated patients, as we wanted to understand the extent of antibiotics versus other medicines being dispensed in this rural province, whether antibiotics were being dispensed with or without a prescription, the perceived indications for antibiotic use, whether with a prescription or not, and the rationale behind any purchasing of antibiotics without a prescription. This compares with the relative rigidity of simulated patients depicting only one or two indications and typically only mentioning specific antibiotics [64,65,66,69,96].

### 4.2. Target Population and Study Sample

Community pharmacies in South Africa are principally categorised into three groups, with the majority of community pharmacies in this province being independent pharmacies, followed by franchise and chain pharmacies (Table 10) [60]. The target population included patients who visited community pharmacies across the province after being dispensed their medicines, whether OTC, dispensed following a prescription, or purchased without a prescription. Considering a target population of 900,106 adults accessing private healthcare in the province [95], and assuming a response distribution of 50% to give the largest sample size at 90% power and 95% confidence level, the minimum recommended sample for a descriptive survey was calculated at 385 (Epi Info version 7.2.4.2; centres for Disease Control and Prevention, Atlanta, United States of America). This minimum target sample size was subsequently increased by 10% to 420 patients to allow for any incomplete data.

Patients were eligible if they were 18 years or older, coming out of the pharmacy with at least one medication, and willing to participate in the survey [58,59]. Considering the feasibility, resources, and practical implications, and given the landscape of community pharmacies in this province, the sampling strategy was to conveniently select patients for an interview as they were exiting community pharmacies, while aiming to reflect the proportional representation of pharmacy categories across the province. This approach was considered more practical and feasible than interviewing potential patients in other locations where suitable numbers of possible interviewees may be low, and there could be recall bias if it had been some time since they had left a community pharmacy. The intention of this approach was to enhance the contextual validity of the findings as well as ensure necessary coverage of varying regulatory and operational frameworks common to these outlets in both rural and peri-urban communities. The targeted patient sample size, distributed by pharmacy category, is shown in Table 10.

### 4.3. Patient Questionnaire Development

The data collection instrument (Appendix A) was specifically designed for the purpose of this study. It was based on previous publications combined with input from key personnel in this area, including leading academic personnel [58,97,98,99,100]. The variables in the initial questionnaire included the extent of medicines being purchased or dispensed, including OTC medicines and antibiotics; the nature of antibiotics being dispensed and their indication, as well as the extent of purchasing antibiotics without a prescription, and the rationale for this. Alongside this, patients’ knowledge of and attitudes towards antibiotics and AMR (not reported in this paper).

Pilot studies had previously been conducted to evaluate the suitability of the initial data collection instrument, which has been described in detail elsewhere [58,59]. The focus of the pilot testing was on key issues, including the clarity, relevance, and the effectiveness of the survey questions and methodologies. The pilot questionnaire was interviewer-administered, followed by direct feedback from participating patients, with patients’ comments subsequently used to recommend potential pertinent modifications to the questionnaire. Potential shortcomings and challenges encountered during the pilot studies were subsequently addressed [58,59]. The goal was to ensure that the instrument used in the main study could successfully elicit the necessary information to meet the study objectives. This combined approach is similar to other published studies in this area among LMICs [49,101,102,103,104,105,106].

The finalised questionnaire was divided into two parts (Appendix A). Part 1 collected information on patients’ sex, their educational level, the nature of the community pharmacy they exited from, the extent of medicines dispensed or sold, whether these included an antibiotic, whether the antibiotics were dispensed with a prescription or not, and the perceived indication or condition for which treatment was sought. Details of any antibiotics dispensed, whether with a prescription or not, were based on their physical examination by the data collectors and subsequently categorised during data analysis by their WHO AWaRe classification into Access, Watch and Reserve antibiotics [21,107].

Potential indications for antibiotic dispensing were based on previous publications and the pilot studies with both community pharmacy personnel and patients. They included URTIs, STIs, UTIs, and skin and soft tissue infections [34,36,55,59,63,98], covering both dispensed and purchased antibiotics. Following this, the rationale for purchasing antibiotics without a prescription, whether any other treatment was suggested or recommended before an antibiotic was suggested by community pharmacy personnel, as well as the key reasons for choosing to approach community pharmacies for their infectious disease in the first place.

The finalised questionnaire was subsequently translated into three commonly spoken local languages, namely Sepedi, Tshivenda, and Xitsonga. The objective was to help with patient understanding of key terms and issues, especially where there were no direct terms for antibiotics and AMR in the local language [59]. This step involved linguistic experts fluent in both English and the three local languages used in the pilot study, i.e., Sepedi, Tshivenda, and Xitsonga, as well as very familiar with the terminology [58,59]. This is similar to the approach of Mokoena et al. (2021), where the study questionnaire was translated into IsiZulu and Setswana, as these were the local languages spoken most frequently by the surveyed taxi community [98].

### 4.4. Patient Recruitment and Data Collection

Three trained research assistants conducted face-to-face interviews with patients after they exited community pharmacies in the province. Data collectors positioned themselves in close proximity to the pharmacy entrances and subsequently approached individuals who were visibly carrying a medicine bag or pharmacy shopping bag. Patients were approached completely independent from pharmacists and pharmacist assistants who were participating in a separate survey as part of the larger mixed-methods study, which was being conducted at approximately the same period of time for comparative purposes [60,94]. Interviews were subsequently conducted only with patients who had just been dispensed medicines, whether OTC or not, preserving the inclusion criteria. Data collectors remained in the vicinity for approximately two hours before moving to a different community pharmacy to enable comprehensive coverage of the different pharmacy types.

As previously mentioned, patients coming out of targeted community pharmacies were conveniently selected, with those who did not appear to have collected any medication or who declined to engage being excluded from the study. Before enrolment, the objectives of the study were clearly explained to each potential participant. An information sheet about the study was read to them using their preferred language commonly spoken in the province, i.e., English, Xitsonga, Tshivenda, and Sepedi, followed by an opportunity to ask any questions they might have about the study. Upon agreement to participate, a written informed consent, including details about the right to participate or withdraw from the study, was completed by each of the participants prior to the interview. Consent forms were also available in English and the most common languages spoken in the province. Similarly, all interviews were conducted in the participant’s preferred language, which was either English, Xitsonga, Tshivenda, or Sepedi. No personal identifiers were collected (Appendix A); consequently, anonymity and confidentiality were maintained for all participants.

### 4.5. Data Management and Analysis

The data entry process was conducted using Microsoft Excel™, where the principal researcher (TM) initially entered all data. Following this, a research assistant performed a double-check of the data entry to ensure its accuracy. Before analysis, the data were cleaned and coded to standardise responses and eliminate any inconsistencies.

Once finalised, the cleaned and coded data were imported into Jamovi 2.7.6 (https://www.jamovi.org/) for descriptive and inferential statistical analysis. Continuous data were summarised using means with standard deviation (SD) and medians with interquartile range (IQR), while frequencies and percentages were calculated for categorical variables with 95% confidence interval (CI) where relevant. Antibiotics were categorised according to the WHO AWaRe classification [21,107]. Chi-square tests were used to ascertain whether there were any significant differences between variables, with *p* < 0.05 considered statistically significant and Cramer’s V calculated to determine the strength of the association. The following associations were investigated:The extent of purchasing of antibiotics without a prescription between different pharmacy categories. Our hypothesis was that there would be differences between pharmacy categories, based on the study of Mokwele et al. and the pilot studies (Table 1).The types of antibiotics and indications for antibiotics dispensed with or without a prescription. Our hypothesis was that there would be a difference based on published studies across LMICs, as well as the existing findings in South Africa (Table 1 and Appendix A).Antibiotics dispensed with or without a prescription from the different AWaRe categories, with a particular focus on Access antibiotics.

## 5. Conclusions

This study offers a detailed examination of antibiotic prescribing and dispensing patterns in this rural South African province, which highlights important AMS activities required going forward. Overall, there was substantial use of Watch antibiotics, including their prescribing for self-limiting infections, as well as dispensing of antibiotics without a prescription for patients with STIs.

We believe the findings provide actionable insights that support AMS efforts across key stakeholder groups, including encouraging greater AWaRe-ness. Alongside this, there is a greater need for monitoring current dispensing practices among community pharmacies, along with upgrading and integrating IT systems across pharmacies to be able to better monitor current dispensing against agreed AWaRe guidance. In addition, exploring further the opportunities for trained community pharmacists in South Africa to be able to dispense antibiotics without a prescription for patients with UTIs and other agreed indications, building on successful experiences in other countries. There is also a need to address the rationale behind the current high rates of dispensing of antibiotics without a prescription for STIs. This could lead to the instigation of pertinent prevention strategies involving community pharmacists and pharmacist assistants, building on published studies in this area.

## Figures and Tables

**Table 1 antibiotics-14-01273-t001:** Extent of purchasing of antibiotics without a prescription in South Africa.

Study	Key Findings
Anstey Watkins et al., 2019 [56]	All 60 community members surveyed in this rural setting knew that it was not permitted to purchase antibiotics without a prescription Most people relied on being provided antibiotics free at the point of access in PHC clinics, with high levels of unemployment in this rural setting However, some patients admitted to keeping them for the next episode of a household infectious disease, either to be used by themselves or shared with other family members or neighbours
Do et al., 2021 [57]	Only 1·2% of 418 purchases/dispensing of antibiotics were provided without a prescription in this rural South African province
Mokwele et al., 2022 [54]	Among 34 community pharmacies visited by SPs, antibiotics were sold without a prescription in 80% of cases in privately owned pharmacies Antibiotics were sold for patients with UTIs (with a fear of being perceived as having STIs), with none for SPs presenting with URTIs No antibiotics were dispensed without a prescription to SPs in corporate (franchised) pharmacies
Sono et al., 2024 and 2025 [58,59]	There was appreciable dispensing of antibiotics in pilot studies where patients were surveyed regarding antibiotic use when exiting community pharmacies in a rural province: ○ A total of 3 out of 5 patients dispensed an antibiotic were dispensed an antibiotic without a prescription—all 3 patients exiting independent pharmacies ○ A total of 11 out of 15 patients were dispensed antibiotics, including 8 without a prescription—all from independent pharmacies (8/10) STIs were the most common indication for dispensing antibiotics without a prescription; appreciably lower for URTIs
Maluleke et al., 2025 [60]	A total of 69.3% of 128 participating pharmacies throughout this South African rural province admitted to dispensing antibiotics without a prescription at some stage during the past 14 days. This was principally among independent as opposed to chain or franchise pharmacies (98.7%) However, estimates provided by participating community pharmacists and assistants suggested that only 8.6% of the total volume of antibiotics dispensed among the 88 community pharmacies admitting to this practice were dispensed without a prescription 97.6% of pharmacist assistants and 98.1% of pharmacists in community pharmacies indicated they always, or mostly, offered symptomatic relief first before dispensing any antibiotics without a prescription to patients presenting with self-limiting conditions such as URTIs

NB: PHC = primary healthcare, SPs = simulated patients, STI = sexually transmitted infection, UTI = urinary tract infection, and URTI = upper respiratory tract infection.

**Table 2 antibiotics-14-01273-t002:** Sociodemographic characteristics of patients interviewed on exit from the different community pharmacy categories.

Pharmacy Category and Sociodemographic Characteristics		Number (%) of Participants (*N* = 465)
Pharmacy category	Chain	80 (17.2)
	Franchise	116 (24.9)
	Independent	269 (57.8)
Sex	Male	232 (49.9)
	Female	233 (50.1)
Language	Tshivenda	50 (10.8)
	Xitsonga	69 (14.8)
	Sepedi	133 (28.6)
	English	213 (45.8)
Education level	None	2 (0.4)
	Secondary school completed	56 (12.0)
	ABET certificate	5 (1.1)
	College certificate	76 (16.3)
	Diploma	145 (31.2)
	Honours	5 (1.1)
	Bachelor’s degree	145 (31.2)
	Master’s degree	27 (5.8)
	Doctoral degree	4 (0.9)

**Table 3 antibiotics-14-01273-t003:** Medicine items dispensed with or without a prescription among surveyed patients from community pharmacies across the province.

Medicine Items	Patients Dispensed Medicine Items		
	With a Prescription (*n* = 127)	Without a Prescription, Including OTC Medicines (*n* = 338)	Total (*N* = 465)
Number (%) of items	369 (34.2)	710 (65.8)	1079
Mean (SD) number of items per patient	2.9 (1.36)	2.1 (0.87)	2.3 (1.09)
Median (IQR) number of items per patient	3.0 (2.0)	2.0 (2.0)	2.0 (1.0)
Minimum number of items per patient	1.0	1.0	1.0
Maximum number of items per patient	6.0	5.0	6.0

NB: OTC = over the counter for minor ailments; SD = standard deviation; IQR = interquartile range.

**Table 4 antibiotics-14-01273-t004:** Antibiotics dispensed to patients with or without a prescription from community pharmacies across the province.

Antibiotics Dispensed		Number (%) of Patients Who Received an Antibiotic		
		With a Prescription *	Without a Prescription *	Total **
Number of antibiotic items	1	42 (31.8)	90 (68.2)	132 (52.2)
	2	12 (10.2)	106 (89.8)	118 (46.6)
	3	0	3 (100)	3 (0.2)
Antibiotic intended for	Adult	33 (19.0)	141 (81.0)	174 (68.8)
	Child #	21 (26.6)	58 (73.4)	79 (31.2)
Total number (% [95%CI])		54 (21.3 [16.7–26.8])	199 (78.7 [73.2–83.3])	253

NB: * row percentages; ** column percentages; 95% CI = 95% confidence interval; # child means that the antibiotics, whether prescribed or dispensed without a prescription, were for a child but collected by an adult.

**Table 5 antibiotics-14-01273-t005:** Antibiotics dispensed with or without a prescription distributed by their AWaRe category.

Antibiotic	AWaRe Category	Number (%) of Antibiotics Dispensed		
		With a Prescription *	Without a Prescription *	Total **
Metronidazole	Access	4 (2.9)	132 (97.1)	136 (36.2)
Azithromycin	Watch	19 (15.4)	104 (84.6)	123 (32.7)
Cotrimoxazole	Access	1 (3.6)	27 (96.4)	28 (7.4)
Co-amoxiclav	Access	18 (85.7)	4 (19.0)	22 (5.9)
Cephalexin	Access	4 (28.6)	10 (71.4)	14 (3.7)
Chloramphenicol	Access	0 (0.0)	14 (100)	14 (3.7)
Flucloxacillin	Access	6 (66.7)	3 (33.3)	9 (2.4)
Cefixime	Watch	2 (28.6)	5 (71.4)	7 (1.9)
Ciprofloxacin	Watch	0 (0.0)	4 (100)	4 (1.1)
Nitrofurantoin	Access	4 (100)	0 (0.0)	4 (1.1)
Doxycycline	Access	0 (0.0)	3 (100)	3 (0.8)
Fucidin	Watch	0 (0.0)	3 (100)	3 (0.8)
Amoxycillin	Access	2 (100)	0 (0.0)	2 (0.5)
Cefpodoxime	Watch	2 (100)	0 (0.0)	2 (0.5)
Erythromycin	Watch	0 (0.0)	2 (100)	2 (0.5)
Fosfomycin	Watch	2 (100)	0 (0.0)	2 (0.5)
Clarithromycin	Watch	1 (100)	0 (0.0)	1 (0.3)
Metronidazole gel	NA	1 (100)	0 (0.0)	1 (0.3)
Total		66 (17.6)	311 (82.7)	377

NB: * row percentages; ** column percentages. AWaRe—Access, Watch and Reserve [21].

**Table 6 antibiotics-14-01273-t006:** Antibiotic items dispensed with or without a prescription and indications/conditions for which patients sought treatment.

Indication/Condition	Number (%) of Antibiotics Dispensed		
	With a Prescription *	Without a Prescription *	Total **
Sexually transmitted infections	1 (0.4)	225 (99.6)	226 (60.1)
Skin and soft tissue infection	9 (14.3)	54 (85.7)	63 (16.8)
Vaginal thrush	0 (0.0)	13 (100.0)	13 (13.5)
Upper respiratory tract infection	39 (92.9)	3 (7.1)	42 (11.2)
Urinary tract infection	9 (64.3)	5 (35.7)	14 (3.7)
Diarrhoea	0 (0.0)	8 (100)	8 (2.1)
Toothache	2 (50.0)	2 (50.0)	4 (1.1)
H. pylori	3 (100.0)	0 (0.0)	3 (0.8)
Dental infections	2 (100.0)	0 (0.0)	2 (0.5)
Eye infection	0 (0.0)	1 (100.0)	1 (0.3)
Total	65 (17.3)	311 (82.7)	377

NB: * row percentages; ** column percentages.

**Table 7 antibiotics-14-01273-t007:** Patients’ reasons for choosing the particular pharmacy for collecting their medicines.

Reason for Choosing the Pharmacy	Number (%) of Patients		
	Received Medicines, Including Antibiotics, on Prescription or OTC *	Received an Antibiotic Without a Prescription *	Total **
Convenience	106 (71.6)	42 (28.4)	148 (31.8)
Familiar with staff	4 (3.5)	111 (96.5)	115 (24.7)
Easy parking	60 (88.2)	8 (11.8)	68 (14.6)
No strict policy	22 (45.8)	26 (54.2)	48 (10.3)
Affordability	27 (96.4)	1 (3.6)	28 (6.0)
Accessibility	20 (90.9)	2 (9.1)	22 (4.7)
Customer service	13 (68.4)	6 (31.6)	19 (4.1)
Friendly staff	7 (70.0)	3 (30.0)	10 (2.2)
Medicine availability	6 (100)	0 (0.0)	6 (1.3)
Trust and reputation	1 (100)	0 (0.0)	1 (0.2)
Total number of patients	266	199	465

NB: * row percentages; ** column percentages.

**Table 8 antibiotics-14-01273-t008:** Patients’ reasons for obtaining antibiotics without a prescription.

Reason	Number (%) of Patients
Used the same antibiotic(s) before	113 (56.8)
Long waiting times	31 (15.6)
No money to consult with a medical practitioner	12 (6.0)
No medicines at the clinic (needed to purchase at pharmacy)	11 (5.5)
The pharmacist recommended them	9 (4.5)
Cheap or affordable	7 (3.5)
The patient insisted on an antibiotic	6 (3.0)
Potential reasons listed are not applicable	5 (2.5)
Clinic too far	3 (1.5)
No medical practitioner around	2 (1.0)
Total	199

**Table 9 antibiotics-14-01273-t009:** Stakeholder-specific recommendations to reduce antibiotic use without a prescription in rural South Africa and across the country.

Stakeholder Group	Recommendations
**National and Provincial Health Authorities**	Urgently enhance the focus on AMR and the National Action Plan (NAP) given rising concerns with AMR in South Africa, exacerbated by increasing use of Watch and Reserve antibiotics [29,31]. There have been concerns with the implementation of NAP across Africa, including South Africa [83,84,85]. This needs to change with the instigation of regular reviews of the NAP, especially now that it is due for an update [29,31] As part of this, the following is recommended: ○ Avoid blanket enforcement of prescription-only access for antibiotics across South Africa, especially given limited dispensing of antibiotics without a prescription for URTIs compared with prescribing habits. Alongside this, concerns with access to prescribers in PHC clinics, potentially appreciable travel costs, and long waiting times ○ Evaluating the potential for trained community pharmacists to prescribe and dispense selected antibiotics for patients with targeted infectious diseases, including STIs and UTIs, building on successful experiences in other countries (Appendix A) and in Ghana [65]. This will involve improved training of community pharmacists and pharmacist assistants, especially regarding antibiotics, AMR, and AMS, building on current initiatives [86,87] ○ Address concerns through education and other initiatives with the current prescribing of antibiotics in PHC clinics. Appendix A provides exemplars going forward. This involves the education of HCPs, including prescribers and dispensers, along with patients, on appropriate antibiotic use, building on the WHO AWaRe classification and guidance [21,80,81] ○ As part of this, make sure all trainee HCPs, including pharmacist assistants, are conversant with the WHO AWaRe classification and guidance on graduation and post-qualification, given current concerns [21,37,80,81,88]. Alongside this, improve the communication skills of trainee pharmacists and pharmacist assistants regarding the possible prevention and management of STIs among patients to reduce repeat antibiotics [70,72] ○ Expand diagnostic capacity in both PHC clinics and community pharmacies, e.g., rapid STI tests, to reduce inappropriate syndromic overuse of antibiotics, especially those from the Watch list, given current concerns ○ Expand ASPs among community pharmacists, pharmacist assistants, and patients (Appendix A). This should include adapting current STGs where necessary, as well as the instigation of agreed quality indicators among community pharmacists based on the increasing use of the WHO AWaRe system for monitoring antibiotic use and guidance [87,89,90]. However, this will require improvements in IT systems and integration across all pharmacy types, along with active monitoring of current utilisation patterns
**District Health Management Teams**	Strengthen district AMS committees to oversee antibiotic use across clinics and pharmacies. As part of this, instigate regular audits and feedback loops, especially regarding dispensing practices and the rationale for this activity. As part of this, install advanced IT systems into community pharmacies where necessary to improve the monitoring of antibiotic dispensing practices alongside instigating and monitoring agreed quality indicators [90]. Integrate pharmacy-level data into district AMR surveillance dashboards to improve future monitoring as well as monitoring possible shortages. This can include improving the uptake and use of the Stock Visibility System (SVS) and RxSolutions more effectively through improved training. Such activities will help build a centralised view of current antibiotic use trends within provinces and nationally Alongside this, community pharmacists receive anonymized peer comparisons and stewardship updates, fostering professional pride and self-regulation, especially following the instigation of agreed quality indicators, to improve antibiotic use in primary care [87]
**Public Healthcare Clinics**	Explore the potential for including community pharmacy personnel within PHC clinics to help treat patients with infectious diseases, to reduce the patient burden on PHC personnel, given current concerns. This has worked well in other countries [91,92] and builds on very limited dispensing of antibiotics without a prescription for patients with URTIs in this and other studies [54,59,60] compared with concerns with antibiotic prescribing generally in South Africa (Appendix A) Promote guideline-based prescribing aligned with STGs and the AWaRe system via instigating appropriate ASPs and monitoring future dispensing, including against agreed quality indicators. Community pharmacists can assist in this regard
**Pharmacists and Pharmacy Assistants**	Encourage pharmacists and their assistants to enrol for the South African Society of Clinical Pharmacy’s (SASOCP) free or subsidised online AMS courses, where pertinent, which can be completed on mobile devices Health authorities, universities, and SASOCP should also advocate for AMS training to be included in CPD workshops or district-level continuing education sessions. Alongside this, trained pharmacists should be encouraged to lead short in-house sessions for colleagues The South African Pharmacy Council should lobby for the ability of trained pharmacists to prescribe an agreed list of antibiotics for agreed conditions at PHC level, given concerns with current prescribing habits in South Africa, alongside concerns with current access and costs to PHC clinics (Supplementary Tables S1). This can build on current initiatives in South Africa as well as other countries (Appendix A) [86]. However, this will mean expanding the existing supplementary Primary Care Drug Therapy training for pharmacists, with currently only one university (North West University) offering this course [86,87] Apply the “4 Ds” of stewardship when dispensing antibiotics, e.g., right drug, dose, duration, and possible de-escalation as part of ASPs building on adapted WHO AWaRe guidance [80,81] Continue to encourage alternatives to antibiotics for self-limiting conditions such as URTIs and engage in discussions regarding the prevention and appropriate management for patients presenting with STIs [85,87]. This can be via visual aids, including posters, flipcharts, or short videos in waiting areas, explaining viral vs. bacterial infections Partner with local schools, churches, and local radio stations to share messages about responsible antibiotic use
**Patients and Communities**	Launch public campaigns on AMR and responsible antibiotic use, building on Antibiotic Guardian programmes [93] Promote understanding of prescription requirements and risks of misuse, as well as prevention programmes where concerns, e.g., for repeated STIs, where antibiotics are being requested Encourage early clinic visits and discourage self-medication by training pharmacists and pharmacist assistants to ask patients about their symptoms and recommend visits to PHC clinics when there are concerns, e.g., prolonged fever or chest pains Provide leaflets or posters in pharmacies
**Universities and Training Institutions**	Critically evaluate current HCP undergraduate curricula to cover key aspects of antibiotics, including the AWaRe system and guidance, AMR, AMS, and ASPs, including principles of ASPs. This starts with a review of current curricula to see if this is fit for purpose [37,87] Promote interprofessional education on syndromic management and stewardship ethics, as well as increase the number of courses available to community pharmacists to be able to dispense an agreed list of antibiotics for agreed conditions [86,87] (Appendix A) Collaborate with district health management teams to host quarterly CPD sessions that bring together nurses, pharmacists, clinical associates, and doctors Assist with the evaluation of any ASP activities in community pharmacies and among patients to improve future antibiotic use, as well as any Government moves to allow community pharmacists to dispense a selected list of antibiotics for targeted infectious diseases (Appendix A). This could include activities surrounding developing, implementing, and monitoring agreed quality indicators based on the WHO AWaRe system and guidance [21,80,81,90] Support rural placement programmes to improve retention and contextual training. This includes modules on syndromic management, AMS, ASPs, and ethical decision-making in low-resource settings. Use real cases from the local clinic or pharmacy to teach practical skills Encourage community engagement projects, e.g., through school health talks and radio segments

NB: AMR = antimicrobial stewardship, AMR = antimicrobial resistance, ASP = antimicrobial stewardship programme, AWaRe = Access, Watch and Reserve [21], CPD = continuing professional development, HCPs = healthcare professionals, PHC = primary healthcare, STI = sexually transmitted infection, UTI = urinary tract infection, URTI = upper respiratory tract infection, WHO = World Health Organization.

**Table 10 antibiotics-14-01273-t010:** Categories of community pharmacies in a rural South African province (adapted from [60]) and targeted sample of patients for recruitment.

Pharmacy Category	Pharmacy Category Description	Number * (%) of Pharmacies	Proportional Targeted Number (%) of Patients for Recruitment
Chain pharmacies	Owned by corporate entities, such as Clicks and Dischem. These operate under centralised systems and branding.	38 (14.0%)	59 (14.0%)
Franchise pharmacies	Independently owned by franchisees but operating under a common brand name. Examples include Link and The Local Choice.	71 (26.1%)	110 (26.2%)
Independent pharmacies	Standalone entities with no corporate or franchise affiliations. These vary in size, clientele, and business models.	163 (59.9%)	251 (59.8%)
Total		272	420

NB: * numbers based on the publicly accessible pharmacy register at the time of the study (https://interns.pharma.mm3.co.za/SearchRegister, accessed on 30 October 2025). Six pharmacies in the province were non-operational at the time of the study, hence not included in the target population.

## Data Availability

Additional data are available from the corresponding author on reasonable request.

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
