# Peer review of "Prevalence and Associated Factors for Purchasing Antibiotics Without a Prescription Among Patients in Rural South Africa: Implications for Addressing Antimicrobial Resistance"

_antibiotics, 2025, doi:10.3390/antibiotics14121273_

Round 1

Reviewer 1 Report

Comments and Suggestions for Authors

An article that highlights the misuse of antibiotics and reveals possible unnecessary treatments for infections that are not bacterial in nature. The article provides important information and raises red flags; however, there are several aspects that need clarification, as follows:

  • why did the authors choose convenience sampling, excluding those who refused the survey or did not purchase medications? Convenience sampling rather than random sampling may increase the prevalence of non-prescription purchases because others were not included.
  • to standardize the statistical results, I ask the authors to add confidence intervals.
  • there are a few minor issues regarding language quality, such as sentences that could be simplified, e.g., “patients coming out of targeted community pharmacies were conveniently selected,” and some long sentences in the discussion section.

Author Response

Comments and Suggestions for Authors

1) An article that highlights the misuse of antibiotics and reveals possible unnecessary treatments for infections that are not bacterial in nature. The article provides important information and raises red flags; however, there are several aspects that need clarification

Author comments: Thank you for these kind words – appreciated! We hope we adequately address your suggestions

The suggestions area as follows:

  • Why did the authors choose convenience sampling, excluding those who refused the survey or did not purchase medications? Convenience sampling rather than random sampling may increase the prevalence of non-prescription purchases because others were not included.

Author comments:  Thank you for this observation. The primary aim of our study was to assess the prevalence of antibiotic use among patients, and subsequently determine whether these antibiotics were prescribed or dispensed without a prescription, including their nature, indications, and rationale. To achieve this, we considered it most appropriate to approach patients directly as they exited community pharmacies with a medicine or pharmacy bag. This ensured that data collection was contemporaneous and minimized recall bias, which could have arisen if patients were asked to report medicines obtained earlier from other settings (e.g., grocery or general stores). We used convenience sampling as this approach provided a practical and reliable means of capturing real-time dispensing practices in the community pharmacy context, which was the specific focus of our study. This also ensured that as many patients as possible could be interviewed, as all pharmacies are not equally busy, which also explains why only a certain period of time was spent the vicinity of each of the pharmacies. Excluding patients who refused to participate in the survey is a required ethical practice, as voluntary participation should be ensured. We hope this clarification addresses the concern.

  • To standardize the statistical results, I ask the authors to add confidence intervals.

Author comments: Thank you for this comment. We have added 95% CI to Tabes 2 and 3 where we describe our overall sample. We do not consider 95% CI necessary for the other tables as we mainly used descriptive statistics and presented proportions from our own sample dataset, thus not subject to uncertainty. Instead, where we did investigate associations, we now also included the strength of the association (practical significance) to provide better context and interpretation for our findings. Calculation of OR with 95% CI was possible for only set of variables, which we now also included. We trust this is now OK.

  • There are a few minor issues regarding language quality, such as sentences that could be simplified, e.g., “patients coming out of targeted community pharmacies were conveniently selected,” and some long sentences in the discussion section.

Author comments: Thank you – we have now gone through the manuscript and updated with the help of co-authors who are native English speakers with over 600 papers published in peer-reviewed Journals between them We trust this is now OK.

Reviewer 2 Report

Comments and Suggestions for Authors

Overall, the paper is very good. The topic is worthy of investigation and addressed an important public health issue. I have 2 minor comments for further improvement.

First of all, the title of the study is too long and it needs to be more precise. I suggest “Prevalence and Associated Factors of Non-prescription Antibiotic Use in Rural South Africa: Implications for Antimicrobial Resistance”.

From table 11, it is better to remove “Other”, when exact reason is mentioned already.

Author Response

Comments and Suggestions for Authors

Overall, the paper is very good. The topic is worthy of investigation and addressed an important public health issue. I have 2 minor comments for further improvement.

Author comments: Thank you for these

1) First of all, the title of the study is too long and it needs to be more precise. I suggest “Prevalence and Associated Factors of Non-prescription Antibiotic Use in Rural South Africa: Implications for Antimicrobial Resistance”.

Author comments: Thank you – now updated. We hope this is now OK.

2) From table 11, it is better to remove “Other”, when exact reason is mentioned already.

Author comments: Thank you – now updated.

Reviewer 3 Report

Comments and Suggestions for Authors

In the manuscript titled "Appreciable purchasing of antibiotics without a prescription among patients in rural South Africa, the rationale and the implications for addressing antimicrobial resistance", the authors used a questionnaire to determine whether or not people were getting antibiotics without a medical prescription and why. As one of the causes of the antimicrobial resistance crisis is the overuse and misuse of antibiotics, understanding why people get antibiotics without a prescription is pivotal to implementing effective local public health policies. Although the study is restricted to a rural area in South Africa, it is so well executed and discussed that it will appeal to Frontiers' global readership, especially from other low- and middle-income countries.  

General comments
The Abstract needs to be rewritten in proper English, as some phrases are incomplete (the verb is missing) or awkwardly worded, making it really hard to understand. In stark contrast, the manuscript is well written and easy to follow.   

Tables 1 and 2 are confusing to me. The information in Table 1 could be easily summarized and incorporated into the paper's introduction, or, better, moved to the discussion. Typically, the introduction of a paper is used to set the basic background of the research, to point out what has been done recently, and the knowledge gaps the current work aims to close. Therefore, the information in Table 2 is appropriate for the introduction, just not in a table format. Alternatively, the data from both tables could be used to inform the paper's discussion. 

Please make sure you define all your numerous abbreviations upon first use. For instance, the term "URTI" appears 18 times in the document, including twice in the Abstract, yet it is only defined in a footnote in Table 2. 

In lines 320 – 321, this phrase seems to lack a verb: "Consequently, a key area for consideration among the authorities in South Africa struggling to provide comprehensive care to patients".

There are many tables in this manuscript. Some could be merged, like Table 7 and Table 9. 

Discussion is rich yet very long. Authors should be more concise. Same with Table 12, which is an important table, but the text could be further summarized. 

Author Response

Comments and Suggestions for Authors

In the manuscript titled "Appreciable purchasing of antibiotics without a prescription among patients in rural South Africa, the rationale and the implications for addressing antimicrobial resistance", the authors used a questionnaire to determine whether or not people were getting antibiotics without a medical prescription and why. As one of the causes of the antimicrobial resistance crisis is the overuse and misuse of antibiotics, understanding why people get antibiotics without a prescription is pivotal to implementing effective local public health policies. Although the study is restricted to a rural area in South Africa, it is so well executed and discussed that it will appeal to Frontiers' global readership, especially from other low- and middle-income countries.  

Author comments: Thank you for these kind words – appreciated! We hope we have now adequately addressed your comments and concerns.

General comments

1) The Abstract needs to be rewritten in proper English, as some phrases are incomplete (the verb is missing) or awkwardly worded, making it really hard to understand. In stark contrast, the manuscript is well written and easy to follow.   

Author comments: Thank you – now hopefully addressed within the word allowance.

2) Tables 1 and 2 are confusing to me. The information in Table 1 could be easily summarized and incorporated into the paper's introduction, or, better, moved to the discussion. Typically, the introduction of a paper is used to set the basic background of the research, to point out what has been done recently, and the knowledge gaps the current work aims to close. Therefore, the information in Table 2 is appropriate for the introduction, just not in a table format. Alternatively, the data from both tables could be used to inform the paper's discussion. 

Author comments: Thank you for this. We have now moved old Table 1 to Supplementary Material. However, we would like to keep Table 2 if we can so Readers can rapidly assimilate the current variable situation in South Africa. We hope this is OK with you.

3) Please make sure you define all your numerous abbreviations upon first use. For instance, the term "URTI" appears 18 times in the document, including twice in the Abstract, yet it is only defined in a footnote in Table 2. 

Author comments: Thank you – now addressed.

4) In lines 320 – 321, this phrase seems to lack a verb: "Consequently, a key area for consideration among the authorities in South Africa struggling to provide comprehensive care to patients".

Author comments: Thank you – now removed as we looked to shorten the Discussion

5) There are many tables in this manuscript. Some could be merged, like Table 7 and Table 9. 

Author comments: Thank you for this. As seen, we have now cut down the number of Tables in the Results section where we can and moved the rest to Supplementary Material. We hope this is now acceptable.

6) Discussion is rich yet very long. Authors should be more concise. Same with Table 12, which is an important table, but the text could be further summarized. 

Author comments: Thank you for this comment. We have now amended the Discussion and the Table where we can, and hope this is now acceptable.

Reviewer 4 Report

Comments and Suggestions for Authors

Dear Authors

The paper examines how often antibiotics are dispensed—especially without a prescription—in community pharmacies in a rural province of South Africa. The study highlights a high rate of non-prescription dispensing, particularly for suspected sexually transmitted infections, and identifies common drivers such as prior antibiotic use and long clinic waiting times. These findings underline the need to review pharmacists’ roles and strengthen antibiotic stewardship.

Thank you for submitting this important and well-prepared manuscript. The study offers meaningful insights into antibiotic dispensing patterns and highlights key areas for improving stewardship efforts. Please adjust the layout so that Tables appear entirely on one page for better readability and easier data interpretation.

 I am pleased to inform you that I recommend acceptance of your paper in its current form.

Author Response

Comments and Suggestions for Authors

Dear Authors

1) The paper examines how often antibiotics are dispensed—especially without a prescription—in community pharmacies in a rural province of South Africa. The study highlights a high rate of non-prescription dispensing, particularly for suspected sexually transmitted infections, and identifies common drivers such as prior antibiotic use and long clinic waiting times. These findings underline the need to review pharmacists’ roles and strengthen antibiotic stewardship.

Thank you for submitting this important and well-prepared manuscript. The study offers meaningful insights into antibiotic dispensing patterns and highlights key areas for improving stewardship efforts. 

Author comments: Thank you for these kind comments – appreciated!

2) Please adjust the layout so that Tables appear entirely on one page for better readability and easier data interpretation.

Author comments: Thank you for this – we will work with the Journal regarding any revised lay-out.

3) I am pleased to inform you that I recommend acceptance of your paper in its current form.

Author comments: Thank you – appreciated!

Round 2

Reviewer 3 Report

Comments and Suggestions for Authors

Thank you for adjusting the manuscript to my suggestions. I have no further comments.